# PET/CT May Assist in Avoiding Pointless Thyroidectomy in Indeterminate Thyroid Nodules: A Narrative Review

**DOI:** 10.3390/cancers15051547

**Published:** 2023-02-28

**Authors:** Gaby Abou Karam, Ajay Malhotra

**Affiliations:** 1Section of Neuroradiology, Department of Radiology and Biomedical Imaging, Yale School of Medicine, 333 Cedar St., New Haven, CT 06510, USA; 2Department of Radiology and Biomedical Imaging, Yale School of Medicine, 789 Howard Ave, New Haven, CT 06519, USA

**Keywords:** indeterminate thyroid nodule, ^18^F-FDG PET/CT, ^18^F-FCH PET/CT, PET metrics, radiomics, cost-effectiveness

## Abstract

**Simple Summary:**

Patients with indeterminate thyroid nodules are overtreated with futile surgery. Even though current guidelines recommend surgery as a first step after identification of an indeterminate thyroid nodule; other alternatives exist, such as PET/CT scan. This narrative review focuses on the major results and the limitations found in most recent studies on PET/CT efficacy (from PET/CT visual assessment to quantitative PET parameters, radiomic features analysis, and predictive models based on several features) and on PET/CT cost-effectiveness compared to alternatives (such as surgery and molecular testing).

**Abstract:**

Indeterminate thyroid nodules (ITN) are commonly encountered among the general population, with a malignancy rate of 10 to 40%. However, many patients may be overtreated with futile surgery for benign ITN. To avoid unnecessary surgery, PET/CT scan is a possible alternative to help differentiate between benign and malignant ITN. In this narrative review, the major results and limitations of the most recent studies on PET/CT efficacy (from PET/CT visual assessment to quantitative PET parameters and recent radiomic features analysis) and on cost-effectiveness (compared to other alternatives (such as surgery)) are presented. PET/CT can reduce futile surgery with visual assessment (around 40%; if ITN ≥ 10 mm). Moreover, PET/CT conventional parameters and radiomic features extracted from PET/CT imaging can be associated together in a predictive model to rule out malignancy in ITN, with a high NPV (96%) when certain criteria are met. Even though promising results were obtained in these recent PET/CT studies, further studies are needed to enable PET/CT to become the definitive diagnostic tool once a thyroid nodule is identified as indeterminate.

## 1. Introduction

Thyroid nodule prevalence depends on the evaluation tool used. In the general population, 4 to 7% and 20 to 70% have a thyroid nodule detected on palpation and on ultrasound assessment, respectively [1,2]. Following thyroid nodule detection by palpation, or incidentally on imaging, guidelines recommend evaluating thyroid nodules for possible malignancy status. The first step in the evaluation process should be measuring TSH and visualizing the thyroid nodule on ultrasound (US). If the TSH level and US features are consistent with a suspicious thyroid nodule (for example, if they include microcalcifications, hypoechogenicity, irregular borders, taller-than-wide shape, the absence of a simple cyst and spongiform nodule, or extension to adjacent tissues and organs), then the next best step should be a fine-needle aspiration biopsy (FNAB) to assess the cytological characteristics of the thyroid nodule and rule out malignancy [3,4,5].

On average, approximately 10 to 30% of thyroid nodules undergoing FNAB are classified as indeterminate on cytological examination [6,7]. Some pathologists consider indeterminate thyroid nodules (ITN) to be thyroid nodules with either Bethesda category III, IV or V [8], while others believe that ITN should only include Bethesda III and IV due to the high rate of malignancy of Bethesda V (50–75%) [7,9].

Bethesda III includes thyroid nodules classified as atypia of undetermined significance or follicular lesions of undetermined significance (AUS/FLUS), while Bethesda IV includes thyroid nodules indicative of a follicular neoplasm (FN/SFN) or a Hürthle cell neoplasm (HCN/SHCN). Bethesda V is composed of thyroid nodules suspicious of malignancy, such as papillary carcinoma, medullary carcinoma, metastatic carcinoma and lymphoma [9]. The reported malignancy rate of ITN ranges from 10–40%, with an approximated risk of 10 to 30% for Bethesda III and 25 to 40% for Bethesda IV [9,10]. In this study, ITN is defined only as thyroid nodules with a Bethesda category of III or IV. Once FNAB is repeated for Bethesda III only, then the next step in ITN management would be to either actively monitor or surgically remove the nodule. In fact, the treatment plan will depend on the nodule’s clinical and imaging characteristics and patient preference [4]. Nonetheless, an important number of patients still undergo futile surgery. Recently, to reduce unnecessary surgery, other tools have been suggested, such as molecular testing or PET/CT scan. In some cases, molecular testing was able to prevent futile surgery, but it has limitations (e.g., lack of long-term studies, accessibility, high cost) [11,12] that have led researchers to review other more affordable and widely available alternatives, such as PET/CT scan.

Currently, in contrast to US and FNAB, PET-CT is not recommended in the initial diagnostic work-up of ITN due to the findings of past studies that analyzed the impact and importance of PET/CT in the diagnosis process of thyroid nodules [3]. However, with the promising development of radiomics, the use of PET/CT in the diagnostic process for thyroid nodules could increase.

In this review, we present a summary of the recent literature evaluating PET/CT, including PET/CT visual assessment efficacy and utility in the initial management of ITN relative to other imaging techniques. We review the efficacy of using PET quantitative metrics to differentiate ITN histopathological status. The cost-effectiveness of PET/CT compared to alternative diagnostic tools is also reviewed. Evidence of the ability of radiomics analysis based on PET/CT imaging and its consequent model to discriminate between benign and malignant ITN is summarized. Finally, we review relevant PET/CT radiomic features (RFs) and models used to assess thyroid incidentaloma that might be extrapolated to contribute to ITN status identification.

## 2. Visual ^18^F-Fluorodeoxyglucose PET/CT Assessment Efficacy

A recent double-blinded randomized controlled multicenter trial demonstrated that ^18^F-fluorodeoxyglucose (FDG) PET/CT scan can decrease by half unnecessary surgery for ITN (with a size ≥ 10 mm). This study included 132 patients who were divided randomly into two groups. The first group was the ^18^F-FDG PET/CT group. It included 91 patients with ITN (69%), all of whom initially had a ^18^F-FDG PET/CT scan. Then patients with a positive FDG uptake were recommended to have surgery, while those with a negative FDG uptake were recommended to have active medical follow-up for their ITN. A second group was composed of 41 patients with ITN (31%) who had undergone surgery. The study results showed, with a *p*-value <0.001, that 42% of patients in the first group compared to 83% in the second group had undergone futile surgery. Moreover, this study demonstrated that ^18^F-FDG PET/CT scan led to a 48% reduction (23/48) in surgery for non-Hürthle cell nodules compared to a 13% (2/15) reduction in Hürthle cell nodules. These results are explained by the fact that Hürthle cell nodules readily take up FDG regardless of their histopathological nature (benign or malignant) [13]. This study also showed that ^18^F-FDG PET/CT had a high sensitivity 94.1%, a high negative predictive value (NPV) 95.1%, and a benign call rate of 31.1%, with low specificity and a positive predictive value (PPV) [13] (Table 1).

The findings of this recent study are consistent with those of previous studies [14,15,16,17,18] that also demonstrated high sensitivity and high NPV of ^18^F-FDG PET/CT when evaluating ITN (with a size ≥ 10 mm). In other words, ^18^F-FDG PET/CT can rule out malignancy of ITN (≥10 mm) with a reasonable degree of certainty when FDG uptake is negative [14,15,16,17].

**Table 1 cancers-15-01547-t001:** Studies reporting PET/CT scan performance in identifying thyroid nodule histopathological characteristics.

Radiotracer	Study	Study Design	TN Size (mm)	Number of Patients	Bethesda Classification	Time of Imaging after Radiotracer Injection	Sensitivity (%)	Specificity (%)	PPV (%)	NPV (%)	Accuracy (%)	Malignant Rate (% (ratio))
^18^F-FDG	De Koster et al. [13]	Prospective	≥10	91	Bethesda III-IV	60 min	94.1	39.8	35.2	95.1	N/A	31 (28/91)
	Piccardo et al. [16]	Prospective	>10	87	TIR3A, TIR3B *	50 min	94	58	37	98	66	21 (18/87)
^18^F-FCH	Ciappuccini et al. [19]	Prospective	≥15	107	Bethesda III-IV-V	20 min	90	50	29	96	55	19 (20/107) **
						60 min	85	49	28	94	67	
				94	Bethesda III-IV	20 min	100	47	17	100	N/A	11 (10/94) **

FDG: fluorodeoxyglucose; FCH: fluorocholine, N/A: not applicable, NPV: negative predictive value, PPV: positive predictive value, TN: thyroid nodule. * TIR3A and TIR3B are equivalent to Bethesda III and IV, respectively. ** Represent the pre-malignant and malignant rate.

## 3. Visual Assessment from PET/CT Compared to Other Imaging Techniques

Piccardo et al. [16] compared the ability of three different imaging techniques to detect malignancy of ITN. This prospective study included 87 patients with ITN (nodules with TIR3A and TIR3B were included, which are the equivalent of Bethesda III and IV, respectively) who had 99mTc-methoxyisobutylisonitrile scintigraphy (99mTc-MIBI-scan), multiparametric neck ultrasonography (MPUS) and ^18^F-FDG PET/CT scan within one day of each other before thyroidectomy for ITN. They showed that ^18^F-FDG PET/CT had significantly better sensitivity, accuracy, and NPV than MPUS and 99mTC-MIBI scan. Moreover, in line with previous studies, the absence of FDG uptake on PET/CT correlated with benign ITN, with a high NPV of 98% [13,14,17,18].

In addition, the study found a specificity of 94% (*p*-value = 0.0001) for detecting thyroid malignancy when an ITN had a positive FDG uptake associated with a negative MIBI. Moreover, in a multivariate analysis (adjusted by age, thyroglobulin levels and nodule dimensions), the association of a positive ^18^F-FDG on PET/CT scan and a negative 99Tc-MIBI scan supported malignancy. In a univariate analysis, positive FDG PET/CT associated with a positive MPUS was associated with a higher specificity for differentiating thyroid carcinoma than the specificity observed for FDG PET/CT on its own.

## 4. ^18^F-Fluorocholine an Alternative to ^18^F-FDG Radiotracer

^18^F-FDG is the radiotracer for PET/CT scan that is routinely used to identify or rule out malignant thyroid nodules, while ^18^F-fluorocholine (FCH) or ^11^C-choline is typically used in prostate cancer. Some patients with prostate cancer who had had a ^18^F-FCH PET/CT scan incidentally showed positive ^18^F-FCH uptake within their thyroid gland that later turned out to be secondary to either a benign or malignant thyroid nodule [20,21,22,23].

These reported cases led Ciappuccini et al. [19] to study the performance of ^18^F-fluorocholine in identifying premalignant (non-invasive follicular thyroid neoplasm (NIFTP)) and malignant ITN. This prospective study included 107 patients with an ITN ≥ 15mm. In this study, Bethesda V nodules were considered as ITN and the population studied included 13, 81 and 13 patients with Bethesda III, IV and V, respectively. The participants underwent ^18^F-fluorocholine PET/CT scan at 20 and 60 min after injection of the radiotracer (1.5 MBq/Kg). Most of the PET/CT test characteristics were slightly better at 20 min than at 60 min with a higher sensitivity and NPV at 20 min with only the accuracy value better at 60 min (Table 1).

This study demonstrated a possible reduction of 48% (*p* < 0.001) in futile surgery by relying on ^18^F-FCH PET/CT to identify benign thyroid nodules among Bethesda III-IV nodules. This reduction was possible because of a high NPV (96% at 20 min).

However, this study had two major limitations. First, Bethesda category V was considered as an ITN, and the authors included 13 patients with a Bethesda V nodule (with 10 malignant nodules). If these patients are removed from the cohort, both the sensitivity and NPV increase to 100% while the specificity remains practically unchanged at 47% and the PPV decreases to 17% at 20 min. The second issue is the possible overestimation of the NPV due to a low premalignancy/malignancy rate of 11% (one NIFTP and nine malignant nodules among the 94 patients with Bethesda III-IV nodules), and 19% (20/107) when Bethesda III-IV-V nodules were included.

Based on the Ciappucini et al. [19] study, it seems that the ^18^F-FCH radiotracer performance is better, or at least similar to, the ^18^F-FDG radiotracer performance when used on patients to detect ITN. Nevertheless, it may be too early to draw this conclusion because more studies are needed to validate these results.

Additionally, ^18^F-FCH has two advantages compared to ^18^F-FDG. Firstly, a lower irradiation dose is needed when the PET/CT radiotracer is ^18^F-FCH (1.5MBq/kg) compared to ^18^F-FDG (usually around 3.7 MBq/kg) [19,24,25]. Secondly, a lower latency time is needed between radiotracer injection and image acquisition for ^18^F-FCH (only 20 min compared to +/− 60 min for ^18^F-FDG) (Table 1). Another consideration when comparing the two radiotracers is their cost. In Europe, both radiotracers have practically the same price according to Ciappucini et al. [19] study, but in other countries their costs might differ.

## 5. PET Quantitative Parameters

A study undertaken by De Koster et al. [26] assessed the ability of quantitative measurements obtained from ^18^F-FDG PET/CT imaging to differentiate preoperative ITN properties. It included 123 patients (55 were Bethesda III nodules, while 68 were Bethesda IV nodules (39 FN/SFN and 29 HCN/SHCN). In this study, PET conventional parameters were measured (mainly SUV_max_, SUV_peak_, SUV_max_-ratio, and SUV_peak_-ratio). A higher median value for conventional parameters was present in malignant/borderline nodules compared to benign nodules, with *p* <0.001 (Table 2). Moreover, similar cut-off values for the conventional parameters were found in non-Hürthle cell nodule groups and all the 123 nodule groups, while a higher cut-off value was measured in the HCN/SHCN group. These cut-off values were associated with high sensitivity in each group (Table 2).

Other studies have also focused on traditional quantitative ^18^F-FDGPET parameters (especially SUV_max_) to determinate ITN histological characteristics. Some have reported a significant correlation between the SUVmax value and ITN benign/malignant status. In fact, it was reported that malignant ITN had a higher SUVmax compared to benign ITN [27]. Similar results were reported when an ^18^F-FCH radiotracer was used instead of ^18^F-FDG [19]. Some reports have suggested a cut-off value to differentiate between benign and malignant ITN [17,28]. Tumoral tissue usually has higher metabolic activity (which leads to a higher SUV_max_) than benign tissue. Nonetheless, SUV_max_ was not always reported as a discriminative tool able to differentiate between them. For example, Nguyen et al. [29] prospectively followed 108 patients with follicular neoplasm or atypia and reported a difference in the median SUV_max_ between malignant ITN (7.2 g/mL) and benign ITN (4.9 g/mL), but this was not statistically significant, with a *p* value = 0.10.

Even though some studies have suggested relying on SUV_max_ as a diagnostic tool to detect malignant ITN, currently, SUV_max_ and other SUV parameters should not be used alone to differentiate between benign/malignant ITN due to the presence of overlapping values between benign and malignant ITN [30] and the inclusion of Hürthle cell nodules within the ITN categories. It is well-known that Hürthle cell nodules have a higher FDG avidity (secondary to the abundance of mitochondria within Hürthle cells) leading to a higher SUV value even if they are benign nodules [26,31,32,33]. Similarly, when an ^18^F-FCH radiotracer was used, a higher SUV_max_ was found in Hürthle cell adenoma and carcinoma compared to other benign and malignant subtypes, respectively [19].

## 6. Cost-Effectiveness of PET/CT Scan

Different pathways exist once an ITN is identified. The classical pathway recommended by current guidelines is diagnostic surgery. However, recently, many patients before undergoing surgery have decided, with the assistance of their physician, to undergo a molecular test (different tests exist) to rule in or out malignant ITN nodules to avoid futile surgery (molecular tests are especially frequent in the United States). Another alternative to molecular testing is ^18^F-FDG PET/CT, which is also not currently recommended as a matter of course before diagnostic surgery for ITN, even though some studies suggest a considerable reduction in futile surgery [13].

A study by Vriens et al. [12] compared the cost-effectiveness of these different pathways and their respective impacts on patient quality of life. They created a Markov decision model based on probabilistic analysis of a 5-year follow-up to compare the cost and effectiveness of the four different pathways (i.e., surgery, gene expression classifier (GEC), mutation marker panel (MMP), ^18^F-FDG PET/CT scan) once a thyroid nodule is characterized as indeterminate on cytology. After a 5-year follow-up, the mean cost of the FDG PET/CT pathway was the lowest compared to the other pathways. Furthermore, only the genetic pathway had a minimally better health-related quality of life outcome than FDG PET/CT due to a higher percentage of futile surgery undergone in the ^18^F-FDG PET/CT group (40%) compared to the GEC group (38%). Moreover, according to the Vriens study, if an ^18^F-FDG PET/CT scan was required in the United State for ITN before surgery, it could theoretically reduce the annual cost by EUR 164 million (which corresponds to approximately USD 177 million (if the current euro-dollar exchange rate is used (15 January 2023: EUR 1.0000 = USD 1.0823) [34])). This study was conducted within the Dutch health system. Nevertheless, the use of an ^18^F-FDG PET/CT scan could play a major role in regions outside the United States where genetic testing might not be available or, if available, is very expensive.

De Koster et al. [35], in a prospective multicentered study, analyzed the cost-effectiveness of using an ^18^F-FDG PET/CT scan compared to surgery in patients with ITN at one year of follow-up. The study included 132 patients with ITN divided into an ^18^F-FDG PET/CT group and surgery group which were followed for 1 year. A total of 106 patients underwent diagnostic surgery during the 1 year of observation with inclusion of crossover between the management pathways in the analysis. At 1 year of follow-up, a mean healthcare cost difference of EUR 1300/patient (*p* = 0.01) was found between the two management strategies in favor of the ^18^F-FDG PET/CT strategy (but this difference became statistically non-significant and decreased to EUR 1000/patient (*p* = 0.06) in favor of ^18^F-FDG PET/CT when healthcare costs related to incidental FDG PET/CT findings were added). A Markov decision model was built to predict the difference in total societal cost (which included all medical costs (not only those limited to thyroid-nodule-related care), costs secondary to productivity losses, and patient costs (such as travel expenses)) between both management strategies over a lifelong period. Even though a mean lifelong societal cost difference of EUR 9900/patient was found, it was not statistically significant, with a *p*-value = 0.14. Regarding the quality of life, no statistically significant difference between both management strategies was found at 1 year follow-up and over the lifelong interval. The previously mentioned study of De Koster et al. [13] found a decrease of approximately 40% in futile surgery when ^18^F-FDG PET/CT was used in ITN before surgery. This decrease in unnecessary surgery, added to the absence of significative differences in cost and quality of life between both management strategies, tends to favor ^18^F-FDG PET/CT over surgery (from a cost-effectiveness point of view).

## 7. Radiomics Based on ^18^F-FDG PET/CT Imaging in Indeterminate Thyroid Nodules

Radiomics has the potential to significantly improve cancer diagnosis and management. It extracts numerous quantitative features from medical imaging. Some of the RFs extracted can serve as biomarkers in models aimed at identifying malignant tumors, predicting oncological patient clinical prognosis, or identifying genomic mutation status [36,37,38,39].

The radiomics extracted from PET/CT imaging provides information on the anatomical location and metabolic activity of the tumoral and surrounding tissue, which represents a potentially powerful tool [40].

Kim et al. [41], in a retrospective study, enrolled 200 patients with ^18^F-FDG incidentaloma who underwent an FNAB. Among the PET/CT parameters studied (SUV_max_, SUV_mean_, MTV, TLG, heterogeneity factor) only the intratumoral metabolic heterogeneity measured by the heterogeneity factor (a derivate of a volume-threshold function; HF = dV/dT) could predict malignancy within the inconclusive FNAB group (31 patients), which included 12, 6 and 13 patients with Bethesda III, IV and V, respectively. They found a cut-off value for the heterogeneity factor (HF) (HF > 2.751 in favor of malignancy), with a sensitivity of 100%, specificity of 60% and AUC of 0.826 with a *p*-value < 0.0001.

Even though this pilot study had some limitations (a small cohort of patients with inconclusive FNAB results, which included Bethesda category V in the analysis group), the results suggest that, by measuring tumor metabolic heterogeneity, it is possible to differentiate between benign and malignant thyroid nodules with better efficacy than using conventional PET/CT parameters.

These results should lead to further research to closely evaluate tumor heterogeneity, which can be measured at a global and local level by relying on first-order histogram-based features and on second-order grey-level co-occurrence matrix (GLCM) features, respectively.

De Koster et al. [26] assessed the ability of quantitative measurements and radiomics based on ^18^F-FDG PET/CT imaging to differentiate preoperative ITN characteristics in a multicenter study. The study included 123 patients (55 were Bethesda III, while 68 were Bethesda IV (39 FN/SFN; 29 HCN/SHCN). A total of 100 patients had surgery, while the other 23 patients had active monitoring.

Only 84 patients (28 Hürthle cell nodules and 56 non-Hürthle cell nodules) had a positive ^18^F-FDG uptake. These 84 patients were included in the radiomics analysis and had PET conventional parameters measured (SUV_max_, SUV_peak_, SUV_max_ -ratio, SUV_peak_ -ratio, TLG). These patients were divided into a training set with 68 patients and a testing set with 16 patients. Additionally, other subgroups for Hürthle cell nodules (with a training and testing set) and non-Hürthle cell nodules (with a training and testing set) were created.

A total of 107 RFs were retrieved using the PyRadiomics software package (version 2.1.2). An SUV_max_ threshold of 50% was applied when the volume of interest was extracted. The 107 RFs were divided into 18 intensity features, 14 shape features and 75 texture features (5 neighboring grey tone difference matrix (NGTDM), 14 grey-level dependence matrix (GLDM), 16 grey-level size zone matrix (GLSZM), 16 grey-level run length matrix (GLRLM) and 24 GLCM).

Only six parameters (i.e., entropy of the intensity histogram, nodule size, high intensity on PET, variance in area size, total lesions glycolysis (TLG), and small areas with low grey levels) were retained in the training set after RF dimensional reduction (the Kaiser–Meyer–Olkin tests were excellent in all folds (≥0.927)).

The PET/CT model created from these features had a low mean area under the curve (AUC) in the test sets that included all ITN nodules (0.461), only the non-Hürthle cell nodules (0.466) and the Hürthle cell nodule (0.537). In addition, similar results were found in the PET model with an AUC of 0.421 when all patients with ITN of less than 64 voxels/volume of interest (VOI) were excluded from the analysis (18 patients excluded).

The radiomic analysis in this study did not improve the discriminating power of ^18^F-FDGPET/CT in ruling out malignancy among ITN compared to ^18^F-FDGPET/CT visual evaluation or its quantitative analysis. In fact, in this study, even quantitative parameters of ^18^F-FDG PET/CT helped to differentiate between malignant and benign Hürthle cell nodules in a better way (with an AUC > 0.7 in all, non-Hürthle and Hürthle cell nodules) than the testing group in the radiomic analysis.

The failure of this predictive model was not secondary to the size of the ITN because no improvement in AUC was found when ITNs less than 64 voxels/VOI were excluded.

A retrospective study undertaken by Giovanella et al. [37] assessed the possibility of relying on PET conventional features and RFs to determinate the final pathological status of the ITN with a positive FDG uptake. The study included 78 patients with FDG positive ITN (35 and 45 patients with Bethesda III and IV, respectively) that were later resected for definite histological diagnosis. First, 107 RFs were retrieved, which included 18 first-order features, 14 shape-based features and 75 matrix-based features (5 NGTDM, 14 GLDM, 16 GLRLM, 16 GLSZM and 24 GLCM). These features were extracted using the PyRadiomics software package (version 2.2.0).

Only two features (GLCM_Autocorrelation and shape_Sphericity) were found to be non-redundant and capable of predicting ITN malignancy (with an AUC = 0.733). They were obtained after ruling out 65 RFs highly correlated to MTV and/or SUV_max_, then, on the remaining 42 RFs, TLG and TSH, a LASSO (least absolute shrinkage and selection operator) logistic regression was applied. These two features with the ITN cytology (Bethesda category) were integrated into the multiparametric model capable of predicting malignancy in ITN with FDG uptake. This model had three outputs (a score of 0, 1 or 2) depending on the number of positive features. It was better in predicting the malignancy status of ITN when the population studied included only non-Hürthle cell nodules (65 patients) compared to when all types of ITN were included (65 non-Hürthle cell + 13 Hürthle cell nodules). In fact, when only non-Hürthle cell nodules (vs. all types of ITN) were included, a score of 0 represented benign nodules with an NPV of 95% (vs. 96%), and a score of 2 represented a possible malignant nodule with a PPV of 79% (vs. 58%, respectively).

Unfortunately, the PET/CT model in the De Koster et al. [26] study produced inconclusive results (i.e., low AUC in all groups). Nevertheless, future studies should apply the same approach by creating a training and a testing set for non-Hurtle cell nodules and Hürthle cell nodules separately to be able to integrate quantitative parameters in the predictive model. In addition, by segmenting the population studied in this manner, a larger number of participants per study are needed to reach enough patients in each group to obtain statistically significant results.

## 8. Radiomics Analysis Based on ^18^F-FDG PET/CT Imaging in Thyroid Incidentaloma and the Potential Application of the Results on ITN

Studies examining the ability of PET/CT radiomics to determine the malignancy status of thyroid incidentalomas were included in this review because there are currently only two studies that have investigated PET/CT radiomics capacity to discriminate ITN characteristics [26,42] (Table 3).

Sollini et al. [43] retrospectively included in their study 50 patients with a thyroid incidentaloma detected by ^18^F-FDG PET/CT. In this study, all the patients had seven PET conventional parameters and four histogram-based features extracted from PET/CT imaging, but only 28 patients with a large enough region of interest (≥64 voxels) had additional matrices-based features retrieved (11 GLRLM, 11 grey-level zone length matrix (GLZLM), 6 GLCM, 2 neighboring grey-level different matrix (NGLDM)) and 2 shape and size features). A 40% SUV_max_ threshold was applied to extract the region of interest and the RFs were retrieved using the Life Image Features Extraction (LIFEx) program package.

Among these 43 features only seven features extracted were defined as potential predictors of thyroid incidentaloma (TI) benign/malignant status. The seven features were SUV_max_, SUV_std_, TLG, MTV, kurtosis, skewness and GLCM_Correlation. Among the seven features, only skewness had possible predictive power to identify pathological thyroid nodule characteristics with a sensitivity of 69%, specificity of 69%, PPV of 57% and NPV of 81%. In addition, a reciprocal correlation was found between skewness and kurtosis, MTV and TLG, and SUV_max_ and SUV_std_, with AUC values of 0.830, 0.970 and 0967, respectively.

GLCM_Correlation might be helpful in rejecting the possibility of having a malignant nodule due to its high NPV (100%). Furthermore, only compacity (a shape and size-based feature) was able to discriminate between TIR categories, with a *p*-value = 0.03.

This study lacked a validation group to verify the results and it did not present a predictive model to differentiate between benign and malignant incidentaloma.

A study by Aksu et al. [44] used the texture analysis obtained by ^18^F-FDG PET/CT to differentiate thyroid incidentaloma pathological characteristics. The study included 60 patients, the majority being oncological patients (non-thyroid cancer), except for three patients. They were divided into two sets. The first set was the training group, with 42 patients, while the second set was the testing group, with 18 patients. The LIFEx software package was used to retrieve RFs and a threshold of 40% SUV_max_ was applied when the region of interest was drawn.

Six conventional PET metrics (SUV_max_, SUV_mean_, SUV_std_, SUV_min_, SUV_peak_, TLG) were measured from PET/CT imaging and 40 RFs were extracted. The RFs were divided into 5 first-order features, 3 shape-based features and 32 matrix-based features (including 11 GLRLM, 11 GLZLM, 7 GLCM and 3 NGLDM features).

The training set univariate analysis demonstrated a significant difference in all traditional PET metrics, 5 first-order and 16 second-order features (GLCM, NGLDM, GLRLM, GLZLM) between benign and malignant thyroid nodules. Among these features, the grey-level run length matrix—run length non-uniformity feature (GLRLM_RLNU) was the most powerful feature to differentiate between benign and malignant nodules with an NPV of 100%. The median values of GLRLM_RLNU (range value) were 43.2 (20–84.5), 105.2 (24.7–645.7) and 60.7 (20.0–645.7) in the benign group, malignant group and both groups, respectively, with a *p*-value <0.001. Finally, to create a predictive model of ITN malignancy, a correlation analysis was first performed among 18 features, with an AUC superior to 0.7 to avoid overfitting of these features. We found that only two of these features (GLRLM_RLNU and SUV_max_) had a correlation coefficient inferior to 0.6. Then these two features were used in five machine learning algorithms. Among them, the random forest algorithm had the best model accuracy (78.6%) with the highest AUC at 0.849. When this algorithm was applied to the testing set, it had a good accuracy of 77.8%, with an AUC of 0.731.

Unlike the Giovanella et al. [37] study, shape_Sphericity RF was not a determinant factor in discriminating between benign/malignant thyroid nodules. The median value in shape_Sphericity (value range) was 1.055 (0.990–1.160) and 1.025 (0.910–1.160) in the benign and malignant groups, respectively (*p*-value = 0.036).

The small number of participants, and the fact that this study was not limited to ITN, could be reasons for the lack of association between shape_Sphericity and thyroid nodule benign/malignant characteristics. GLCM_autocorrelation RF was not extracted in this study.

Ceriani et al. [45] extracted traditional PET parameters and RFs from 104 patients with 107 FDG positive nodules. This study involved the creation of a multiparametric predictive model based on three parameters (SUV_max_, TLG and shape_Sphericity) that could identify thyroid incidentaloma malignancy status, with a PPV that could reach 100% when these three parameters were positive. The shape_Sphericity parameter was the only predictor of malignancy among six uncorrelated and non-redundant RFs (Shape_Maximum, 2DDiameterSlice, Firstorder_Energy, GLCM_InverseDifferenceMoment, GLCM_Constrast and GLCM_SumSquares) that were identified within the 107 RFs as potential malignancy predictors. The RFs were divided into 18 first-order features, 14 shape-based features, and 75 matrix-based features (including 24 GLCM, 16 GLRLM, 16 GLSZM, 14 GLDM and 5 NGTDM features). The RFs were extracted using the PyRadiomics software package (version 2.2.0). Five PET conventional parameters (SUV_max_, SUV_mean_, SUV_peak_, MTV, TLG) and six RFs were identified as potential predictors of malignancy, so they were included in a multivariate stepwise logistic regression analysis. This analysis showed that only three features (SUV_max_, TLG and shape_Sphericity) were statistically significant, with a *p*-value < 0.0001.

In this study, the mean shape_Sphericity of the benign and malignant lesion was 0.67 (0.53–0.78) and 0.79 (0.72–0.82), respectively, with a *p*-value < 0.0004.

Regarding shape_Sphericity, it is an RF that represents the circularity of a nodule in comparison with a sphere. It seems to be an important malignancy predictor factor because tumoral tissue expands in an unorganized way. This anarchic expansion can be measured by shape_Sphericity. Moreover, two different studies with two different types of population (one with TI [45], the other with only ITN [42]) had shape_Sphericity as one of their predictive factors used in their respective malignancy predictive models.

GLCM_Autocorrelation is another RF that measures how fine or coarse a texture is. It was considered to be a predictor factor of ITN histopathological status in the Giovanella et al. [42] study. However, the Ceriani et al. [45] study was the only one to measure this feature among the three radiomic studies on TI and the authors did not find the GLCM_Autocorrelation feature to be a predictive feature of TI histopathological status. Since TI includes more histological-type nodules this can lead to a more significant variation in texture within each category (i.e., more different histological types of benign and malignant nodules).

In a study conducted by Sollini et al. [43], the inclusion of four PET metrics as potential predictors of TI histopathological status highlighted the importance and the need to include quantitative PET metrics in future models that can predict the histopathological diagnosis of ITN. For this reason, any future PET/CT radiomic study on ITN should divide the cohort into two groups (Hürthle cell nodules and non-Hürthle cell nodules) due to higher radiotracer uptake and quantitative parameters of Hürthle cell compared to non-Hürthle cell nodules. Moreover, the inclusion of PET quantitative parameters in both the predictive models of Aksu et al. [44] (only SUV_max_ ) and Ceriani et al. [45] (SUV_max_ and TLG) highlights the importance of separating Hürthle cell and non-Hürthle cell nodules.

## 9. Conclusions

Although the current guideline only recommends diagnostic surgery for ITN, more patients are choosing to have a PET/CT or a molecular test to avoid surgery. In this review, we presented the results of recent studies on ITN and the different ways PET/CT scan can assess the benign/malignant status of an ITN to reduce unnecessary surgery.

Visual assessment by PET/CT can prevent around 40% of futile surgeries in patients with ITN (Bethesda III-IV) of at least 10 mm. In addition, PET/CT quantitative metrics may be valuable as possible predictors of malignancy in ITN. However, they should not be used on their own to determine the histopathological status of ITN. Instead, they should be integrated into a predictive model. This predictive model should also select specific RFs extracted from PET/CT imaging that are predictors of benign or malignant nodules (especially: shape_Sphericity or GLCM_Autocorrelation as used in the Giovanella et al. [42] study model).

Another important aspect of a diagnostic tool is its cost-effectiveness. Based on current studies, PET/CT may be a more cost-effective tool compared to diagnostic surgery (with a preserved quality of life in patients who chose a PET/CT active follow-up).

Finally, these promising results are paving the way for new studies that could potentially make PET/CT the definitive diagnostic tool once a thyroid nodule is identified as indeterminate.

## 10. Future Directions

To the best of our knowledge, only two studies have used radiomics extracted from ^18^F-FDG PET/CT to create a predictive model able to identify ITN histopathological characteristics. Future studies should divide cohorts into two groups (Hürthle cell nodules and non-Hürthle cell nodules). Each of these groups should have a training and a testing group. Future predictive models should not be limited to radiomic features only (especially shape_Sphericity) but should also include PET quantitative parameters and other clinical factors, such as cytology features, TSH, thyroglobulin, age, and sex. In addition, there may be value in adding RFs extracted from another imaging technique, such as ultrasound imaging.

Finally, regarding the PET/CT radiotracer, ^18^F-FCH seems to be a valid (if not a better) alternative to an ^18^F-FDG radiotracer in PET/CT. Nevertheless, prospective studies on the ^18^F-FCH radiotracer that includes only Bethesda III–IV nodules are needed to confirm the results reported in the Ciappuccini et al. [19] study.

## Figures and Tables

**Table 2 cancers-15-01547-t002:** Quantitative parameter cut-off values and their respective sensitivity in discriminating between malignant and benign thyroid nodules in three different groups (all nodule types, non-Hürthle and Hürthle cell nodules).

	SUV_max_ (g/mL)	SUV_peak_ (g/mL)	SUV_max_-ratio (g/mL)	SUV_peak_ (g/mL)	Sensitivity (%)
All nodule types (*n* = 123)	2.1	1.6	1.2	0.9	97
Non-Hürthle cell nodules group (*n* = 94)	2.1	1.6	1.2	0.9	95.8
Hürthle cell nodules group (*n* = 29)	5.2	4.7	3.4	2.8	100

© 2023 De Koster et al. This table information comes from Table 2 (Threshold analysis and diagnostic accuracy) (only column one, two and seven were used) found in de Koster et al. [26] study (https://doi.org/10.1007/s00259-022-05712-0) and is licensed under **CC BY 4.0** (http://creativecommons.org/licenses/by/4.0/).

**Table 3 cancers-15-01547-t003:** Studies that relied on radiomics to determine thyroid nodule histopathological characteristics.

Type of Thyroid Nodules	Study	Study Design	Number of Patients	Software	RF Extracted	Features Included in the Predictive Model	AUC
ITN studies	De Koster et al. [26]	Prospective	84 (68 train set, 16 test set)	PyRadiomics (version 2.2.1).	107 RF: 18 IF, 14 SF and 75 TF (5 NGTDM, 14 GLDM, 16 GLSZM, 16 GLRLM and 24 GLCM)	Entropy of the intensity histogram, nodule size, high intensity on PET, variance in area size, TLG, small areas with low grey levels	0.461 (all FDG-positive nodules) 0.466 (NHCN) 0.537 (HCN) 0.421 (only ITN ≥ 64 voxel/VOI)
	Giovanella et al. [42]	Retrospective	78	PyRadiomics (version 2.2.0).		GLCM_Autocorrelation and Shape_Sphericity	0.733
Thyroid incidentaloma studies	Sollini et al. [43]	Retrospective	50	LifeX	4 RF for 22 patients (4 IF); 36 RF for 28 patients: 2 shape and size features, 4 IF and 30 TF (2 NGLDM, 11 GLZLM, 11 GLRLM and 6 GLCM)	N/A (only skewness had possible predictive power to identify histopathological thyroid nodule characteristics)	N/A
	Aksu et al. [44]	Retrospective	60 (42 train set, 18 test set)	LifeX	40 RF: 5 IF, 3 SF and 32 TF (3 NGLDM, 11 GLZLM, 11 GLRLM and 7 GLCM)	SUVmax and GLRLM_RLNU	0.849 (train set) 0.731 (test set)
	Ceriani et al. [45]	Retrospective	104 (with 107 nodules)	PyRadiomics (version 2.2.0).	107 RF: 18 IF, 14 SF, and 75 TF (5 NGTDM, 14 GLDM, 16 GLSZM, 16 GLRLM and 24 GLCM)	TLG, SUVmax, and Shape_Sphericity	0.830

AUC: area under the curve, FDG: fluorodeoxyglucose, GLCM: grey-level co-occurrence matrix, GLDM: grey-level dependence matrix, GLRLM: grey-level run length matrix, GLRLM_RLNU: grey-level run length matrix—run length non-uniformity, GLSZM: grey-level size zone matrix, GLZLM: grey-level zone length matrix, HCN: Hürthle cell nodules, IF: intensity features, ITN: indeterminate thyroid nodules, LifeX: life image features extraction, N/A: not applicable, NGLDM: neighboring grey-level different matrix, NGTDM: neighboring grey tone difference matrix, NHCN: non-Hürthle cell nodules, RF: radiomic features, SF: shape features, TF: texture features, TLG: total lesion glycolysis, VOI: volume of interest.

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
