# Peer review of "PET/CT May Assist in Avoiding Pointless Thyroidectomy in Indeterminate Thyroid Nodules: A Narrative Review"

_cancers, 2023, doi:10.3390/cancers15051547_

Round 1

Reviewer 1 Report

Overview

This paper is a review for investigating a role of PET/CT to avoid unnecessary thyroidectomy in indeterminate thyroid nodules (defined as Bethesta category of III or IV in cytology). The authors concluded the visual assessment by FGD PET/CT can prevent 40% of futile surgery for indeterminate thyroid nodules (ITN) because of high sensitivity and negative predictive value (NPV) of this method. The authors also examined cost-effectiveness of PET/CT comparing to the other strategies, such as diagnostic surgery and molecular testing and found that PET/CT is superior to the others. As for PET/CT quantitative metrics and radiomics analysis based on PET/CT imaging, we still don’t have decisive parameters or radiomics features (RFs) currently for determining histological status of ITNs, however shape-sphericity seemed to be promising.

Minor points

1. Typos

P1L26: Even tough>>Even though

Similar typos are found in P4L178, P5L228

P3L140: NIFTI>>NIFTP

P4L152: (table 1)>>(Table 1)

P4L162, L165: Table 3>>Table 2

2. Abbreviation

P7L338: LIFEx program >>Life Imaging Features Extraction (LIFEx) program

This abbreviation is explained afterwards but it appears on this cite at the first time.

Author Response

Point 1: Typos

P1L26: Even tough>>Even though

Similar typos are found in P4L178, P5L228

P3L140: NIFTI>>NIFTP

P4L152: (table 1)>>(Table 1)

P4L162, L165: Table 3>>Table 2

Response 1: We adjusted it.

Point 2: Abbreviation

P7L338: LIFEx program >>Life Imaging Features Extraction (LIFEx) program

This abbreviation is explained afterwards but it appears on this cite at the first time.

Response 2: We adjusted it.

Reviewer 2 Report

The study by Gaby A Karam et.al is well designed and interesting to read, however a few minor issues must be resolved before considering the manuscript for publication.

Line 1 – please avoid using questions in the title.

Line 39 – 40 – what are the features that help to rule out malignancy?

Line 76 – “Body of the Narrative Review” – I do not think that it is necessary to indicate that the main body of the manuscript starts in this line

Line 94, line 130 and others – there are no tables in the manuscript

Line 98 – “In other words, 18F-FDG PET/CT can rule out malignancy of ITN (≥ 10 mm) with a high certainty when FDG uptake is negative” – please provide a reference

Line 168, line 180 – “benign-malignant status” – did you mean benign/malignant status (like in line 340)??

Line 345 – perhaps the data presented in this and neighboring sections could be shown in the form of a table to compare the results between the studies mentioned in these sections?

Line 352 – please don’t start a new line with a number

Line 452, 453 – “(…)and other clinical factors such as cytology nature, TSH, thyroglobulin, etc”. – please avoid using ETC in this line and replace it with the exact names of factors.

Round 2

Reviewer 2 Report

All my commentaries have been replied to. I now think that the manuscript is ready to be considered for publication. I would only suggest to check the manuscript carefully, because of many changes that have been done to it. Please also reconsider changing the title into something more specific, such as: "PET/CT may help to avoid (unnecessary) thyroidectomy..."

Author Response

The title was adjusted as requested.
Plus, some minor changes were made to the syntax.